# Systemic Immune-Inflammation Index: A Novel Predictor of Coronary Thrombus Burden in Patients with Non-ST Acute Coronary Syndrome

**DOI:** 10.3390/medicina58020143

**Published:** 2022-01-18

**Authors:** Uğur Özkan, Muhammet Gürdoğan, Cihan Öztürk, Melik Demir, Ömer Feridun Akkuş, Efe Yılmaz, Servet Altay

**Affiliations:** Department of Cardiology, School of Medicine, Trakya University, Edirne 22030, Turkey; drmgurdogan@gmail.com (M.G.); dr.cihanozturk@gmail.com (C.Ö.); melikdemir34@gmail.com (M.D.); akkusferidun@gmail.com (Ö.F.A.); drefeyilmaz@gmail.com (E.Y.); svtaltay@gmail.com (S.A.)

**Keywords:** systemic immune-inflammation index, coronary thrombus burden, non-st acute coronary syn-drome, atherosclerosis, plaque rupture, thrombus formation, P2Y12 inhibitors, pretreatment

## Abstract

*Background and Objectives:* Excessive coronary thrombus burden is known to cause an increase in mortality and major adverse cardiac events (MACEs) in NSTE-ACS (non-ST acute coronary syndrome) patients. We investigated the association between the systemic immune-inflammation index (SII) and coronary thrombus burden in patients with non-ST segment elevation myocardial infarction (NSTEMI) who underwent coronary angiography and percutaneous coronary intervention (PCI). *Materials and Methods:* A total of 389 patients with the diagnosis of NSTEMI participated in our study. Coronary thrombus burden was classified in the TIMI (thrombolysis in myocardial infarction) thrombus grade scale and patients were divided into two groups: a TIMI thrombus grade 0–1 group (*n* = 209, 157 males) and a TIMI thrombus grade 2–6 group (*n* = 180, 118 males). Demographics, angiographic lesion images, coronary thrombus burden, clinical risk factors, laboratory parameters, and SII score were compared between the two groups. *Results:* The high thrombus burden patient group had a higher neutrophil count, WBC count, platelet count, and systemic immune-inflammation index (SII) (*p* < 0.001). The receiver operating characteristic (ROC) curve analysis showed that at a cutoff of 1103, the value of SII manifested 74.4% sensitivity and 74.6% specificity for detecting a high coronary thrombus burden. *Conclusions:* Our study showed that the SII levels at hospital admission were independently associated with high coronary thrombus with NSTEMI.

## 1. Introduction

Coronary vessel wall inflammation-causing atherosclerosis and cardiovascular diseases (CVD) are the most common causes of mortality and account for approximately 30% of all deaths [1]. They exhibit an extensive scale, varying from stable coronary lesion to acute coronary syndrome (ACS). ACS is a highly fatal and major cardiovascular event (MACE) manifestation of CVD, usually associated with the rupture of atherosclerotic plaque. Approximately 5.5–18.2% of ACS patients die upon hospitalization [2,3] and a massive intracoronary thrombus burden has been reported in 16.4% of patients with ACS [4].

Though atheroma plaque was once associated solely and exclusively with aberrant cholesterol accumulation, today, it is known that atherosclerosis is a dynamic and inflammatory process of the vessel wall, and the contribution of inflammation and leukocytes to all stages of atherosclerosis can be observed [5]. Vulnerable atheroma plaques essentially include high levels of inflammatory cells and a massive lipid core shielded by a thin fibrous cap [6]. The majority of ACS is a result of coronary thrombus which consists of a sudden rupture of this vulnerable plaque-causing platelet aggregation followed by thrombus formation in the major epicardial coronary artery. As the process of coronary thrombosis formation begins, inflammatory markers start to rise; the inflammation process could be followed by the monitoring of these markers in peripheral blood. Except for inflammation, plaque destabilization and rupture can be triggered by various conditions such as metabolic, hemodynamics, neurohumoral, psychosocial stress, smoking, etc., and in addition, all these reasons can trigger inflammation [7]. In addition, when thrombus formation and platelet activation begin, activated platelets produce eicosanoids and platelet-activating factors that have potent effects on inflammation, and these are released when thrombus formation begins. Another factor is the inadequate adaptation of the myocardium to the ischemia that develops as a result of acute thrombus formation, as in stable stenosis. Therefore, inflammation is more extensive in the area affected by ischemia [8].

The amount of coronary thrombosis depends on the actual thrombotic–thrombolytic equilibrium. This equilibrium of thrombosis cascade exhibits a broad spectrum, varying from mural limited thrombus to vascular occlusive thrombus and massive thrombus burden is associated with a higher incidence of periprocedural thrombotic complications and unsuccessful reperfusion. Therefore, especially in this patient group, we need biomarkers to show that the stability of the lesion may deteriorate due to the thrombus load and that will guide us for earlier invasive intervention or pretreatment with P2Y12 receptor inhibitors for the limitation of thrombus formation.

The systemic immune-inflammation index (SII) is one of the inflammation parameters that have been proven to be practical in many diseases like cancers and CVDs [9,10,11,12,13]. SII is calculated by (N × P)/L (N, P, and L represent neutrophil counts, platelet counts, and lymphocyte counts, respectively). The SII represents three important immune response paths: inflammation that is reflected by neutrophilia, thrombosis that is reflected by platelets, and the body’s stress response, reflected by low lymphocyte [14,15]. Nevertheless, the association between SII levels and coronary thrombosis formation has not yet been investigated. In our study, we researched the function of the SII in estimating the risk of excessive coronary thrombosis formation in non-ST ACS (NSTE-ACS) patients who underwent coronary angiography (CAG).

## 2. Method

### 2.1. Study Population

We retrospectively analyzed data from 389 NSTE-ACS patients who underwent percutaneous coronary intervention (PCI) between January 2020 and January 2021.

NSTEMI was diagnosed according to the European Society of Cardiology (ESC) criteria including acute chest discomfort, and a rise in myocardial necrosis biomarkers without permanent ST-segment elevation electrocardiography (ECG). ECG changes may include any T-wave or ST-segment abnormality without ST-segment elevation > 20 min or the ECG may be normal [16]. Considering the results of CAG, significant stenosis was accepted as 50% or more stenosis of main epicardial arteries or their main branches. If significant epicardial coronary lesions were detected in two or more coronary vessels of the patient, the artery with a thrombosed lesion or more critical lesion was thought as the culprit lesion.

We excluded patients with active infection (including COVID-19), patients with a diagnosis of ischemic heart disease, using antiplatelet or anticoagulant drugs before hospitalization, any chronic metabolic inflammatory situations, using immunosuppressive or chronic anti-inflammatory medication, and congestive heart failure history (ejection fraction < 40%). In addition, patients who were in the acute decompensated heart failure clinic or who had refractory ventricular arrhythmias at the time of admission were excluded.

Our study was approved by the Trakya University Medical Faculty Ethics Committee (TUTF-BAEK 2021/263) and complied with the Helsinki Declaration.

### 2.2. Clinical Data Collection

The medical records of 389 NSTE-ACS patients were analyzed in terms of age, gender, vascular disease, smoking, etc. The laboratory analysis involved neutrophil, lymphocyte, platelet, white blood count (WBC), hemoglobin, monocyte, renal and liver tests, lipid profiles, and cardiac biomarkers. Echocardiography was performed on all patients by using the Simpson method and the ejection fraction (EF) was noted (with Vivid S7; GE Medical System) within the first day of admission to the coronary care unit. Lesions in the epicardial arteries were detected after CAG was performed. In addition, hemodynamic parameters were recorded on admission.

### 2.3. Coronary Angiography and Medications

CAG was performed on all patients within 72 h after hospital admission. All patients received dual antiplatelet therapy (acetylsalicylic acid and clopidogrel) before the diagnostic coronary angiography following the established guidelines [17]. It was performed for patient standardization only in patients using clopidogrel. Radial or femoral artery access was used for the angiography depending on the operator’s preference.

Intravenous heparin/low molecular heparin therapy was used for anticoagulation during PCI. The Gp2b3a inhibitor (tirofiban/abciximab) was used during the percutaneous intervention by the decision of the cardiologist. The images of CAG were determined by at least two cardiologists, or a cardiologist and a cardiovascular surgeon, and treatment strategies were selected following the recommendation of guidelines [16].

Thrombolysis in myocardial infarction flow grade (TIMI), TIMI thrombus grade scale, and Gensini score were recorded from the angiographic images. The TIMI is the blood flow scores of infarct-related vessels that range from 0 to 3 [18]. TIMI thrombus grade scale is an index for the cine-angiographic characteristics of thrombus. It is classified from 0 to 6, based on thrombus burden and stenotic lesion [19,20]. The Gensini score is an angiographic scoring system that is used to quantify and characterize the complexity of coronary artery disease (CAD). It is calculated for each lesion by consideration of the severity score, lesion zone, and coronary collateral adjustment factor. The final score is obtained by summing each coronary lesion score [21].

## 3. Statistical Analysis

The data analysis was performed using the SPSS 21.0 version (SPSS Inc., Chicago, IL, USA). A Shapiro–Wilk test was used to control the distribution of quantitative variables. According to the normality of distribution, descriptive data were given as mean + standard deviation and median (interquartile range [IQR]). Median and IQR were given in cases where the variable was not normally distributed. The independent-samples t-test was used to compare normally distributed quantitative variables, and the Mann–Whitney U test was used to compare non-normally distributed quantitative variables. Categorical variables were compared with the chi-squared test. Categorical variables are all shown as percentages or numbers. The effects of different variables on the coronary thrombus burden were calculated with a univariate analysis. For the multivariate regression analysis, parameters with a *p* < 0.05 in the univariate analysis were included in the model. The cutoff level of SII and NLR in predicting the coronary thrombus burden was determined by performing a receiver operating characteristic curve (ROC) analysis. The value corresponding to the highest sensitivity and specificity value in the ROC analysis was accepted as the optimal cutoff value. A 2-sided *p* < 0.05 was considered significant.

## 4. Results

We recorded 389 NSTE-ACS patients who have had CAG in the study. All participants were divided into a TIMI thrombus grade 0–1 group (*n* = 209, 157 males) and a TIMI thrombus grade 2–6 group (*n* = 180, 118 males) according to the angiographic images. The demographic variables are listed in Table 1.

Except for gender, all demographic characteristics were similar between both groups (*p* > 0.05, for all). The number of males was higher in the low thrombus grade group (*p* < 0.05). Additionally, the LVEF and hospitalization days were similar between groups.

The baseline pre-angiographic hospital laboratory parameters are summarized in Table 2.

After the laboratory measurements were investigated, there was no significant variability between the two groups among lipid profiles, transaminase values, albumin levels, and hemoglobin counts (for all, *p* > 0.05). The low thrombus grade group (TIMI thrombus grade 0–1) had higher creatinine levels and lymphocyte counts (*p* < 0.05). Neutrophil count, WBC count, platelet count, and SII were significantly higher in the high thrombus grade group (TIMI thrombus grade 2–6) (*p* < 0.001).

Angiographic data are listed in Table 3.

The low thrombus grade group (TIMI thrombus grade 0–1) had higher LAD lesion and preprocedural TIMI flow grades (*p* < 0.043 and *p* < 0.001, respectively). The high thrombus grade group (TIMI thrombus grade 2–6) had higher use of Gp2b3a medication, stent implantation, stent length, and postprocedural TIMI flow grade (*p* < 0.001).

The effects of various coronary thrombus burden risk factors were investigated by a multivariate analysis (Table 4).

In the univariate logistic regression analysis, neutrophil count, lymphocyte count, PLT count, WBC count, SII, male gender, creatinine level, and culprit vessel were associated with a high coronary thrombus burden (TIMI thrombus grade 2–6). In addition, this group was correlated with using Gp2b3a medication, stent use, and low preprocedural TIMI flow grade.

A multivariate logistic regression analysis indicated that a high SII score was an independent predictor of a high coronary thrombus burden (odds ratio [OR]: 1.003, 95% confidence interval [CI]: 1.002–1.003, *p* < 0.001) together with WBC count (OR: 1.328, 95% CI: 1.190–1.481, *p* < 0.001). The ROC curve analysis confirmed that an optimum value of 1103 for SII predicted the coronary thrombus burden with 74.4% sensitivity and 74.4% specificity (area under ROC curve 0.794 [95% CI: 0.749–0.839], *p* = 0.023). For the neutrophil to lymphocyte ratio (NLR), the optimum value of 7.35 predicted the coronary thrombus burden with a sensitivity of 61.1% and specificity of 60.8%, and the area under the curve was 0.686 (95% CI: 0.633–0.739; *p* = 0.027; Figure 1).

## 5. Discussion

The most remarkable finding of our investigation is that the increase in SII score is the independent predictor of massive coronary thrombus burden in NSTEMI patients undergoing PCI. The SII has been demonstrated as a new inflammation parameter and according to our knowledge, this study is the first research in the literature to exhibit the correlation between SII and coronary thrombus burden.

As a fatal complication of coronary heart disease, ACS has very high morbidity and mortality [1]. Recent studies have demonstrated that atherosclerosis is not only a hypoperfusion state but also associated with a high level of pro-inflammatory molecules and inflammation [5]. Increased pro-inflammatory markers have been associated with atherothrombotic events. Thrombosis plays a crucial function in the pathogenesis of ACS, as the rupture of an atherosclerotic plaque.

Despite successful PCI, a high coronary thrombus load causes increased use of glycoprotein IIb/IIIa (Gp2b/3a) inhibitors, lower TIMI 3 epicardial coronary flow after the procedure, increased rates of the no-reflow phenomenon, and increased mortality and morbidity. In addition, longer stent use, an increase in the rate of recurrent angiography, longer hospital stays, and more health expenditures were observed in these patients.

The CURE trial was the milestone trial that demonstrated the benefits of the addition of the P2Y12 receptor blocker before angiography, showing a reduction in the composite cardiovascular outcome in patients with NSTE-ACS [22]. After the CURE trial, CURRENT-OASIS7 (2010), TRITON-TIMI 38 (2007), and PLATO (2009) trials established the benefits of the P2Y12 receptor blocker pre-treatment before coronary angiography in these patients [23,24,25].

In 2020, the Swedish Coronary Angiography and Angioplasty Registry (SCAAR) disclosed the observational dataset of 64 857 NSTE-ACS patients and showed that, contrary to what is known, pre-treatment with P2Y12 was not associated with the advanced ischemic event, but with increased bleeding complications [26]. After this analysis, ESC removed the recommendation for P2Y12 receptor blocker loading before angiography in NSTE-ACS patients. This recommendation contradicts the situation where we want to prevent the catastrophic worsening of coronary perfusion and achieve lesion stabilization and successful myocardial reperfusion as early as possible. Following the authors’ hypothesis, this occurred due to the pre-treatment of P2Y12 receptor blockers regardless of whether the myocardial infarction was due to a stable atherosclerosis lesion or the development of atherothrombosis. Therefore, we believe that pre-treatment will increase the chance of success if it can be predicted that the myocardial infarction will develop from an atherothrombotic background.

Recent studies have proven that inflammatory cells can be beneficial to estimate ACS patients’ prognosis, but one or two component-based inflammatory predictors are quite poor and insufficient to predict the prognosis in ACS [27,28]. Therefore, the SII includes three inflammatory cell types and can display the inflammatory situation in the body in detail [29,30].

In recent years, many studies have shown that the increase in the SII is related to poor outcomes in cardiovascular diseases. Esenboga et al. concluded that high SII levels in patients undergoing primary intervention for ST-segment elevation myocardial infarction (STEMI) may be an encouraging index to predict the non-reflow phenomenon [31]. Yang et al. found that high SII levels were related to poor clinical outcomes in coronary artery diseases [13]. Dey et al. found a correlation between the high SII and poor outcomes after an elective off-pump coronary artery bypass graft operation. Huang et al. demonstrated that high SII predicts poor clinical outcomes for elderly patients after ACS [12].

In our study, these two groups were compared according to demographic characteristics, laboratory findings, and coronary lesion burdens. Except for gender, all demographic characteristics and cardiovascular risk factors were similar between both groups (*p* > 0.05, for all). In addition, laboratory findings were similar except WBC, neutrophils, lymphocyte, platelet counts, and SII value. The coronary lesion burden between these groups was compared according to the Gensini score and the Gensini score of both groups were similar. The Gensini and TIMI thrombus scoring systems are unbiased and confidential to describe the CAD. When we examined the lesion distribution, we found that the left anterior descending artery (LAD) lesion was higher in the group with a low thrombus burden. EF, Killip classes, and hospitalization days were similar in both groups.

In our study, the estimation capabilities of the index and each factor were compared through a univariate and multivariate Cox regression analysis. We found that the increase in the SII score was the independent predictor of massive coronary thrombus burden in NSTEMI patients at a similar coronary lesion. The ROC curve analysis confirmed an optimum value of 1103 for SII to predict the coronary thrombus burden. When we analyzed the angiographic results of the groups, we saw that the group with the high SII required more and longer stenting for lesion stabilization. We also found that GP2b3a usage was higher.

The main limitation of our study is all participating patients were given ASA and clopidogrel before diagnostic angiography according to the guidelines. Though we know that pretreatment may have a limited thrombus burden, it is still valuable to demonstrate the relationship between SII and thrombus burden. As the first study about SII and coronary thrombus burden, these results we have obtained need to be confirmed further in large-scale studies.

## 6. Conclusions

Our study outcomes suggest that the increase in the SII score is correlated with the increase in the coronary thrombus burden. Therefore, it can be used to identify patients with a high thrombus-related ischemic risk. This may give us an idea of pretreatment with P2Y12 inhibitors before angiography. Future randomized trials are required.

## Figures and Tables

**Figure 1 medicina-58-00143-f001:**
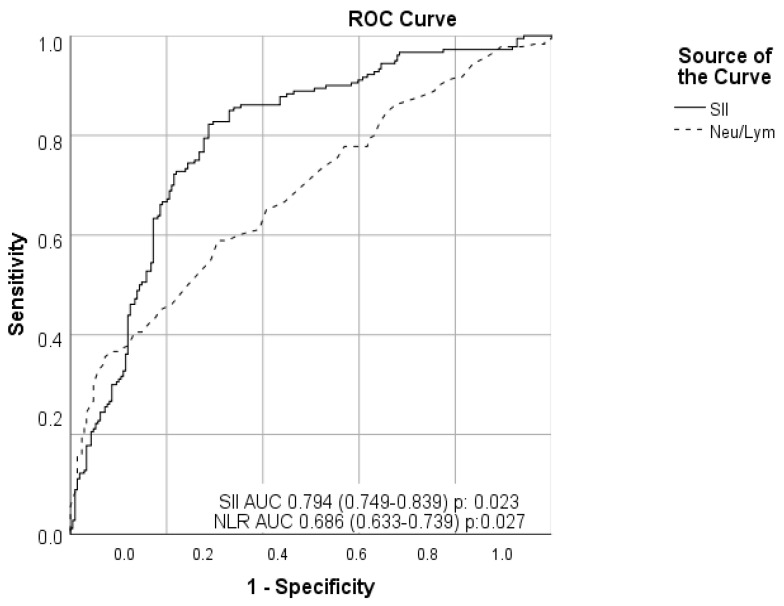
Effect of systemic immune inflammation index and neutrophil–lymphocyte ratio values on coronary thrombus burden (ROC analysis).

**Table 1 medicina-58-00143-t001:** Demographic Characteristics of the Study Populations.

Characteristics	TIMI Thrombus Grade 0–1 (*n* = 209)	TIMI Thrombus Grade 2,3,4,5,6 (*n* = 180)	*p*-Value
Age (years)	66.5 ± 12.6	66.1 ± 13.2	0.77
Male gender, *n* (%)	157 (75.1%)	118 (65.5%)	0.03
Diabetes Mellitus *n* (%)	105 (50.2%)	87 (48.3%)	0.70
Hypertension, *n* (%)	169 (80.8%)	137 (76.1%)	0.25
Stroke, *n* (%)	14 (6.6%)	11 (6.1)	0.81
Smoking, *n* (%)	124 (59.3%)	132 (59.6%)	0.89
Drinking, *n* (%)	49 (23.4%)	40 (22.2%)	0.77
Left ventricular ejection fraction (%)	51.5 ± 5.4	52.1 ± 6.3	0.28
Killip class, 1 2 3	168 (80.3%) 39 (18.6%) 2 (0.95%)	134 (74.4%) 43 (23.8%) 3 (1.6%)	0.35
Hospitalization day	5.85 ± 2.41	5.66 ± 2.21	0.43

**Table 2 medicina-58-00143-t002:** Laboratory Findings of the Study Populations.

Variable	TIMI Thrombus Grade 0–1 (*n* = 209)	TIMI Thrombus Grade 2,3,4,5,6 (*n* = 180)	*p*-Value
Creatinine (mg/dL)	0.98 ± 0.25	0.96 ± 0.29	0.02
AST (µ/L)	45.7 ± 31.9	42.4 ± 30.7	0.3
ALT (µ/L)	27.7 ± 13.1	28.8 ± 18.2	0.47
Albumin (g/dL)	4.04 ± 1.76	3.77 ± 0.44	0.19
Total cholesterol (mg/dL)	197 ± 38	199 ± 39	0.6
HDL-C (mg/dL)	43.3 ± 6.8	42.1 ± 7	0.09
LDL-C (mg/dL)	137 ± 31	139 ± 31	0.58
Triglycerides (mg/dL)	168 ± 69	173 ± 68	0.54
WBC count (10^3^/µL)	9.06 ± 2.31	10.5 ± 2.35	<0.001
Platelet count (10^3^/µL)	264 ± 64	306 ± 79	<0.001
Hemoglobin (mg/dL)	13.1 ± 1.73	12.8 ± 1.46	0.1
Neutrophil count (10^3^/µL)	7 ± 1.52	8.39 ± 2.23	<0.001
Lymphocyte count (10^3^/µL)	2.12 ± 0.63	1.95 ± 0.6	0.008
Monocyte count (10^3^/µL)	0.82 ± 0.48	0.81 ± 0.30	0.76
SII	924 ± 358	1342 ± 399	<0.001

**Table 3 medicina-58-00143-t003:** Angiographic and Procedural Characteristics.

	TIMI Thrombus Grade 0–1 (*n* = 209)	TIMI Thrombus Grade 2,3,4,5,6 (*n* = 180)	*p*-Value
Culprit vessel, *n* (%)			
LAD	123 (58.85%)	83 (46.11%)	0.043
LCX	33 (15.78%)	38 (21.11%)	
RCA	53 (25.35%)	59 (32.77%)	
No. of diseased vessels *n* (%)			
1	59 (28.22%)	54 (30%)	0.55
2	37 (17.70%)	38 (21.11%)	
3	113 (54.06%)	88 (48.88%)	
Gp2b3a use, *n* (%)	13 (6.22%)	55 (30.25%)	<0.001
Stent use, *n* (%)	64 (30.62%)	144 (80%)	<0.001
Stent diameter, mm	2.93 ± 0.42	2.84 ± 0.31	0.07
Stent length, mm	23.32 ± 5.53	27.08 ± 7	<0.001
TIMI thrombus grade scale, *n* (% in all patients)			
0	190 (48.84%)	-	
1	19 (4.88%)	-	
2	-	31 (7.96%)	
3	-	66 (16.96%)	
4	-	56 (14.39)	
5	-	27 (6.94%)	
6	-	0	
Gensini score	43.52 ± 19.75	43.77 ± 20.66	0.903
Preprocedural TIMI grade, *n* (%)			
0	0 (0%)	26 (14.4%)	<0.001
1	2 (0.95%)	70 (38.8%)	
2	205 (98%)	84 (46.6%)	
3	2 (0.95%)	0 (0%)	
Postprocedural TIMI grade, *n* (%)			
0	0 (0%)	0 (0%)	<0.001
1	2 (0.95%)	5 (2.77%)	
2	145 (69.3%)	59 (32.7%)	
3	62 (29.6%)	116 (64.4%)	

**Table 4 medicina-58-00143-t004:** Univariate and Multivariate Predictors of Coronary Thrombus Burden in Patients with Non-ST Segment Elevation.

	Univariate Analysis *			Multivariate Analysis **		
	Odds Ratio	(95% C.I.for Odds Ratio)	*p*	Odds Ratio	(95% C.I.for Odds Ratio)	*p*
Neut	1.491	(1.318–1.687)	<0.001	-	-	-
Lym	0.640	(0.458–0.894)	0.009	-	-	-
PLT	1.008	(1.005–1.011)	<0.001	-	-	-
Stent use	9.062	(5.670–14.485)	<0.001	-	-	-
Gp2b3a use	0.151	(0.079–0.287)	<0.001	-	-	-
SII	1.003	(1.002–1.004)	<0.001	1.003	(1.002–1.003)	<0.001
WBC	1.306	(1.191–1.431)	<0.001	1.328	(1.190–1.481)	<0.001
Gender	0.630	(0.406–0.978)	0.039	-	-	ns.
Creatinine	0.741	(0.357–1.538)	0.421	-	-	ns.
Culprit vessel	1.305	(1.037–1.643)	0.023	-	-	ns.
Preprocedural TIMI	0.012	(0.003–0.048)	<0.001	25.217	(12.739–49.918)	<0.001

* Logistic Regression (Method = Enter), ** Logistic Regression (Method = Backward Stepwise (Wald)), ns.: non-significant.

## Data Availability

The data presented in this study are available on request from the corresponding author.

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
