# Peer review of "Systemic Immune-Inflammation Index: A Novel Predictor of Coronary Thrombus Burden in Patients with Non-ST Acute Coronary Syndrome"

_medicina, 2022, doi:10.3390/medicina58020143_

Round 1
Reviewer 1 Report
The present study investigated the association between systemic immune inflammation index (SII) and coronary thrombus burden in patients with non-ST segment elevation myocardial infarction (NSTEMI) who underwent coronary angiography and percutaneous coronary intervention (PCI).
First of all, due to the poor English quality, the manuscript is hard to follow.
All abbreviations even in the abstract should be defined on their first appearance in the text ( for example "MACE", "NSTE-ACS").
Non-ST Acute Coronary Syndrome
"SII is calculated by multiplying platelet counts to neutrophil counts and then dividing the lymphocyte count." A better idea would be to insert a formula here.
What the authors wanted to say by "antiagregan".
"Our study was observed by the Trakya Universtiy Medical Faculty Ethics Committee". How did this Committed observe the study?
"Laboratory analysis involved the hemogram parameters (such as neutrophils, lymphocyte, platelet), renal and liver tests, lipid profiles and cardiac biomarker." The authors should be more precise and mention the measured parameters.
"In addition, hemodynamic parameters were recorded on admission." Which hemodynamic parameters and where are the results for these parameters?
"In our study, the participants were classified into two groups in relation to the presence (TIMI trombus grade2-6) or absence (TIMI trombus grade 0-1) of angiographically evident thrombus." (lines 240-242) same information was already given at the beginning of the results section.
The Discussion section should be more focused on the results of the present study.
Conclusions do not reflect the results of the present study.
Overall the manuscript does not follow the journal recommendations: https://www.mdpi.com/journal/medicina/instructions.
Author Response
Response to Reviewer 1 Comments
Many thanks for your didactic and comprehensive evaluation of the manuscript. Below, you will find our responses:
Point 1. First of all, due to the poor English quality, the manuscript is hard to follow.
Response 1: As per your suggestion, the article has been revised by a native speaker.
Point 2. All abbreviations even in the abstract should be defined on their first appearance in the text ( for example "MACE", "NSTE-ACS"). Non-ST Acute Coronary Syndrome
Response 2: As per your suggestion, abbreviations has been explained.
Point 3. "SII is calculated by multiplying platelet counts to neutrophil counts and then dividing the lymphocyte count." A better idea would be to insert a formula here.
Response 3: As per your suggestions, SII has been redefined and expressed with the formula.
SII is calculated by (N×P)/L (N, P and L represent neutrophil counts, platelet counts and lymphocyte counts, respectively).
Point 4. What the authors wanted to say by "antiagregan".
Response 4: As per your suggestions, the word "antiagregan" has been revised.
Point 5. "Our study was observed by the Trakya Universtiy Medical Faculty Ethics Committee". How did this Committed observe the study?
Response 5: As per your suggestion, the relevant sentence has been revised.
Our study was approved by the Trakya Universtiy Medical Faculty Ethics Committee (TUTF-BAEK 2021/263) and complied with the Helsinki Declaration.
Point 6. "Laboratory analysis involved the hemogram parameters (such as neutrophils, lymphocyte, platelet), renal and liver tests, lipid profiles and cardiac biomarker." The authors should be more precise and mention the measured parameters.
Response 6: As per your suggestion, the relevant sentence has been revised.
Laboratory analysis involved neutrophils, lymphocyte, platelet, white blood count (WBC), hemoglobin, monocyte, renal and liver tests, lipid profiles and cardiac biomarker.
Point 7. "In addition, hemodynamic parameters were recorded on admission." Which hemodynamic parameters and where are the results for these parameters?
Response 7: As per your suggestion, the relevant sentence has been removed to avoid misunderstanding.
These parameters included blood pressure and heart rhythm. They were used to exclude patients with non-ST acute coronary syndrome in decompensated heart failure clinic or refractory ventricular arrhythmias at admission. Emergency coronary revascularization was performed in this patient group. Therefore, this group was excluded in order to standardize the effect of dual antiplatelet therapy. This information has been added to the exclusion criteria.
“In addition, patients who were in the clinic of acute decompensated heart failure or had refractory ventricular arrhythmias at the time of admission were excluded”.
Point 8. "In our study, the participants were classified into two groups in relation to the presence (TIMI trombus grade2-6) or absence (TIMI trombus grade 0-1) of angiographically evident thrombus." (lines 240-242) same information was already given at the beginning of the results section
Response 8: As per your suggestion, relevent sentence has been removed from the discussion section.
Point 9. The Discussion section should be more focused on the results of the present study.
Response 9: As per your suggestions, discussion section has been revised.
Point 10. Conclusions do not reflect the results of the present study.
Response 10: As per your suggestions, discussion section has been revised.
Point 11. Overall the manuscript does not follow the journal recommendations: https://www.mdpi.com/journal/medicina/instructions.
Response 11: As per your suggestions, the manuscript has been revised in accordance with the journal recommendations.
Reviewer 2 Report
In the paper “Systemic Immune Inflammation Index: A Novel Predictor of Coronary Thrombus Burden in Patients With Non-ST Acute Coronary Syndrome” authors investigate the association between Systemic immune-inflammation index (SII) and the coronary thrombus formation in SCA patients.
The topic is interesting as the result they found. An higher SII is correlated with an higher thrombus burden nevertheless the discussion is inconsistent.
Authors should better focalise on the potential mechanisms involved in the correlation between the SII and thrombus formation. Moreover, the judgement about the use of GP2b3a is out of argument.
Why author decided to choose a p level <0.10 in the multivariate model? This determines a selection bias in the multivariate model.
One limitation of the study is the use of clopidogrel in all patients. Author should explicate why there is a so high frequency of clopidogrel administration in a SCA-NSTEMI population as it is.
An extensive English language review is recommended
Minors spelliong errors to correct:
Line 94: anjiography
Line 95: %50
Line 100: antiagregan
Line 112-113: “The lesions or thrombus in epicardial arteries were enrolled after performing CAG” the sentence must be reformulated, it is not clear.
Line 153: “p > .005” it is probably a typing error instead of “p>0.05”
Line 205: epiardial flow
Line 203: “The excessive coronary thrombus burden to the contrary successful PCI is related with increased in use of glycoprotein IIb/IIIa (Gp2b/3a) inhibitors, lower ratio of postprocedural normal myocardial reperfusions and TIMI 3 epiardial flow, no-reflow phenomenon, longer stent use, recurrent angiography, prolonged hospitalization, increased morbidity and mortality so that increased in health expenditure (21)” the sentence must be reformulated, it is not clear.
Author Response
Response to Reviewer 2 Comments
Many thanks for your didactic and comprehensive evaluation of the manuscript. Below, you will find our responses:
Point 1. Authors should better focalise on the potential mechanisms involved in the correlation between the SII and thrombus formation. Moreover, the judgement about the use of GP2b3a is out of argument.
Response 1: As per your suggestion, the relevant sentence has been added to the İntroduction section.
“Also, when thrombus formation and platelet activation begins, activated platelets produce eicosanoids and platelet-activating factors that have potent effects on inflammation and they are released when thrombus formation begins. Another factor is the inadequate adaptation of the myocardium to the ischemia that develops as a result of acute thrombus formation, as in stable stenosis. Therefore, inflammation is more in the area affected by ischemia (8).”
References section has been revised.
Point 2. Why author decided to choose a p level <0.10 in the multivariate model? This determines a selection bias in the multivariate model.
Response 2: As per your suggestion, misspelling has been corrected. In our study, parameters with p<0.05 in univariate analysis were included in the model for multivariate regression analysis
Point 3. One limitation of the study is the use of clopidogrel in all patients. Author should explicate why there is a so high frequency of clopidogrel administration in a SCA-NSTEMI population as it is.
Response 3: In our study, It was performed for patient standardization only in patients using clopidogrel. Another reason for choosing clopidogrel is non-clopidogrel P2Y12 inhibitors were more potent effect to suppress thrombus formation. The sentence has been added to the Coronary angiography and medications section.
“It was performed for patient standardization only in patients using clopidogrel.”
Point 4. An extensive English language review is recommended
Response 4: As per your suggestion, the article has been revised by a native speaker.
Point 5. Minors spelliong errors to correct
Response 5: As per your suggestion, the article has been revised for spelling errors.
Point 6. “The lesions or thrombus in epicardial arteries were enrolled after performing CAG” the sentence must be reformulated, it is not clear.
Response 6: As per your suggestion, the relevant sentence has been revised.
Lesions in the epicardial arteries were detected after CAG was performed.
Point 7. “The excessive coronary thrombus burden to the contrary successful PCI is related with increased in use of glycoprotein IIb/IIIa (Gp2b/3a) inhibitors, lower ratio of postprocedural normal myocardial reperfusions and TIMI 3 epiardial flow, no-reflow phenomenon, longer stent use, recurrent angiography, prolonged hospitalization, increased morbidity and mortality so that increased in health expenditure (21)” the sentence must be reformulated, it is not clear.
Response 7: As per your suggestion, the relevant sentence has been revised.
“Despite successful PCI, high coronary thrombus load causes increased use of glycoprotein IIb/IIIa (Gp2b/3a) inhibitors, lower TIMI 3 epicardial coronary flow after the procedure, increased rates of the no-reflow phenomenon, and increased mortality and morbidity. In addition, longer stent use, an increase in the rate of recurrent angiography, longer hospital stay, and more health expenditures were observed in these patients.”

Round 2
Reviewer 1 Report
The authors did some improvements to the manuscript. Still, there are no visible improvements for the English revision and even if the authors say that they revised the manuscript in accordance with the journal recommendations references are not in the recommended format.
Reviewer 2 Report
Authors changed the draft accordingly to the requests
thank you for the opportunity to review your paper
Author Response
Point 1. Authors changed the draft accordingly to the requests thank you for the opportunity to review your paper
Response 1: Many thanks for your didactic and comprehensive evaluation of the manuscript. Yours faithfully.